# Tent–PSO-Based Unmanned Aerial Vehicle Path Planning for Cooperative Relay Networks in Dynamic User Environments

**DOI:** 10.3390/s25072005

**Published:** 2025-03-23

**Authors:** Shuyue Liu, Wenmao Zhou, Mingwei Qin, Xin Peng

**Affiliations:** School of Information Engineering, Southwest University of Science and Technology, Mianyang 621010, China; liushuyue@mails.swust.edu.cn (S.L.); wenmaozhou996@163.com (W.Z.); px@mails.swust.edu.cn (X.P.)

**Keywords:** multi-UAVs, coverage path planning, tent, PSO

## Abstract

Unmanned aerial vehicles (UAVs) equipped with communication devices are easy to deploy and can serve as temporary mobile communication stations, covering dynamic ground users and establishing communication links. The proper planning of UAV paths can improve user coverage while maintaining communication quality. The traditional particle swarm optimization (PSO) algorithm exhibits strong global search capabilities; however, it suffers from challenges such as uneven search coverage owing to the random distribution of the initial particle positions. Additionally, improper parameter selection often leads to premature convergence to local optima. This study proposes a method for multi-UAV coverage of dynamic users by combining Tent chaotic mapping with PSO (Tent–PSO). By using Tent chaotic mapping to adjust a UAV’s speed and position, the method ensures that particles are evenly distributed in the search space. Simulation results demonstrate that Tent–PSO improves the coverage of the global optimal solution by 10% and increases throughput by at least 37%, confirming its effectiveness in covering dynamic user scenarios.

## 1. Introduction

Unmanned aerial vehicles (UAVs) can function as mobile relay stations equipped with communication equipment, enable temporary mobile communication base stations, and improve signal coverage and communication quality [1,2]. Relay UAVs offer the advantages of high flexibility and can be deployed at high altitudes, thereby enhancing the probability of line-of-sight (LOS) transmission. Thus, UAVs offer significant benefits to wireless communication systems [3,4,5]. Several dynamic users typically exist in the context of relay UAV wireless communication tasks, and dynamic user mobility causes the operational area of the relay UAVs to change during task execution. Therefore, the service area of a relay UAV must be flexibly adjusted in response to changes in the communication service users [6]. During communication missions, relay UAVs often need to serve dynamically changing communication users, such as people in disaster areas or crowds in complex environments. The number and location of ground users change continuously, leading to dynamic variations in the service areas of relay UAVs [7].

Effective path planning for relay UAVs necessitates obstacle and hazard avoidance while ensuring adherence to operational constraints. Existing relay UAV path-planning methods are categorized into two categories: deep reinforcement learning (DRL) and bio-inspired optimization algorithms.

### 1.1. DRL

DRL is a machine learning method based on artificial neural networks. By constructing multilayer neural networks, it simulates the learning process of the human brain and gradually achieves maximum reward through interaction with the environment [8]. To address the challenge of UAV communication interruptions caused by frequent changes in the channel model between UAV and users in urban environments with many buildings, a neural network-based strategy has been proposed [9,10]. This approach processes environmental information provided as input and signal strength to predict discrete environmental types, allowing the UAV to anticipate dynamic user mobility in the environment and plan its path, thereby improving communication performance. To address the slow response of traditional coverage path planning to dynamic environments, a DRL-based coverage path algorithm was previously proposed [11]. First, the environmental information sensed by the SAR-UAV was modeled, and then the DRL algorithm was used to design the action space and reward function of the UAV based on the environmental model. Finally, deep learning methods were applied to control the UAV flight to complete coverage trajectory planning.

### 1.2. Bio-Inspired Optimization Algorithms

Bio-inspired optimization algorithms are optimization methods based on biomimetics, such as genetic algorithms (GAs), ant colony optimization (ACO), and particle swarm optimization (PSO).

GAs optimize problem solutions and determine the optimal solution by simulating biological mechanisms, such as natural selection, inheritance, crossover, and mutation. However, GAs may involve a computational load when the solution space is large, converging slowly in the early stages. In [12], a novel GA based on matrix operations was proposed, where matrix operations were implicitly used during the selection, crossover, and mutation stages of chromosomes, helping to reduce the algorithm’s computation time. In [13], a hybrid GA and A* algorithm for optimizing energy consumption were proposed. The GA generates many potential paths, and the A* algorithm identifies the optimal path from these candidates.

ACO is an optimization method that simulates the foraging behavior of ants in nature. It solves combinatorial optimization problems by mimicking the collective behavior exhibited by ants searching for food. However, in complex environments, the algorithm tends to converge more slowly. To address these challenges, ref. [14] introduced a bending resistance weight factor to reduce the number of path bends and large cumulative bending angles. In addition, the pheromone expansion factor was used to improve the pheromone update rule, thereby enhancing the convergence speed and global optimality of the algorithm. In [15], a solution combining a greedy allocation strategy and an improved ACO algorithm was proposed. First, the number of UAVs assigned to a task is determined using the greedy allocation strategy. Subsequently, a variable pheromone enhancement factor and variable pheromone evaporation coefficient are introduced to adjust the pheromone update rule in the ACO algorithm, aiming to improve the accuracy of the algorithm’s solution.

PSO has high search efficiency and a certain degree of adaptability. It performs well in solving problems with multiple local optima and converges faster, with fewer parameters to adjust. However, it is sensitive to the initial population values and may easily fall into local optima. Many scholars have attempted to improve PSO. The authors of [16] proposed a new framework using PSO to reduce communication energy consumption between UAVs and users. This framework utilizes improved PSO for UAV path planning to support user rate requirements. First, it uses the LOS probability to determine the optimal target location to ensure LOS communication between the UAV and user. The framework then uses the improved PSO to determine the most energy-efficient path from the source to target. However, this study did not consider UAV path planning when the user was moving. To address the issue of PSO easily falling into local optima, reference [17] combined the simulated annealing (SA) algorithm with PSO to improve its global optimization capability. If the candidate global optimal solution is worse, its acceptance is determined based on the acceptance probability p of the SA algorithm. By allowing the algorithm to accept worse solutions, the global exploration capability is increased in the early stages, thereby preventing PSO from falling into local optima. However, this method complicates the parameter tuning process, which can deteriorate the optimization results. The authors of [18] combined BAS with PSO for UAV path planning. BAS can further optimize the individual and global optimal solutions of particles, helping PSO to accelerate convergence. However, the combination of these two algorithms significantly increases the computational cost.

DRL offers significant advantages in UAV path planning in high-dimensional environments. The algorithm continuously learns from environmental feedback and effectively adapts to dynamic user mobility in complex scenarios. However, this method requires large amounts of training data and computational resources, and the training process is time-consuming. Bio-inspired optimization algorithms have strong global search capabilities, making them suitable for multi-objective optimization problems or global path planning in complex environments. However, the results depend heavily on the parameters of the algorithm. Among these, PSO was chosen in this study for relay UAV path planning because of fewer parameters adjustments, faster convergence speed, and simpler implementation.

### 1.3. Chaotic Map

Chaos exhibits randomness, ergodicity, and sensitivity to initial values, which enables algorithms to converge quickly. The randomness of chaotic systems can provide algorithms with rich search capabilities, helping them escape the local optima. Tent chaotic mapping is a typical mapping in chaotic systems. Its nonlinear characteristics and high sensitivity to initial conditions result in the powerful chaotic behavior of Tent mapping.

This study mainly investigated how to adjust the path of relay UAVs to maximize the coverage of dynamic users while maintaining communication link quality. First, the dynamic user movement path was simulated using the Gaussian–Markov model, and relay UAV dynamic constraints and an environment model were established to simulate a normal relay UAV flight mission environment. Tent chaotic mapping was then used to initialize the velocity and position of the UAV, ensuring that the initial positions of the relay UAVs were evenly distributed within the environment. Finally, Tent chaotic mapping was applied to adaptively adjust the weight coefficient and learning factor, followed by updating the relay UAV’s positions and velocities.

The remainder of this paper is organized as follows: Section 2 presents the environmental model and relay UAV dynamic constraints and proposes the optimization objectives of the algorithm. Section 3 discusses improvements to the proposed PSO algorithm. Section 4 provides the simulation analysis of the results. Finally, Section 5 summarizes the study and presents future research directions.

## 2. System Model and Problem Description

Figure 1 presents the environmental model, where I=(0,1,…,i−1) represents the set of UAVs, and K=(0,1,…,k−1) represents the set of ground terminal users. We assumed that the UAVs’ flight altitude remains constant; therefore, the coverage range of each UAV is fixed, and each UAV can only communicate with users within its own coverage area. The UAV obtains real-time data from the ground via wireless communication and computes the next position of the UAV by considering historical information, current user data, and environmental data.

### 2.1. User Mobility Model

The user mobility model is a mathematical or algorithmic model that describes user movement behavior. It is widely used to study network coverage, resource allocation, and communication optimization in dynamic environments [6]. In UAV communication and coverage optimization, user mobility models can simulate the dynamic user mobility in user positions to evaluate the effectiveness of UAV path planning in a more realistic manner.

The Gaussian–Markov mobility model was selected to simulate the user mobility process [19]. Figure 2 illustrates the process of simulating user movements using this model. The model is used to simulate smooth changes in user positions and velocities during movement. By introducing the Markov model, the user’s velocity and direction at the next time step are updated based on the user’s state at the previous time step. This approach renders user movements more consistent with the actual physical laws. In two-dimensional space, the user’s velocity and direction are updated according to Equations (1) and (2).(1)$$\left\{\begin{array}{c} v_{t}=\alpha v_{t-1}+\left(1-\alpha \right)\times \bar{v}+\sqrt{1-\alpha^{2}}\times G\\ \theta_{t}=\alpha \theta_{t-1}+\left(1-\alpha \right)\times \bar{\theta }+\sqrt{1-\alpha^{2}}\times G \end{array}\right.$$(2)$$\left\{\begin{array}{c} x_{t}=x_{t-1}+v_{t}*\cos\left(\theta_{t}\right)*∆t\\ y_{t}=y_{t-1}+v_{t}*\sin(\theta_{t})*∆t \end{array}\right.$$

G represents a Gaussian random variable, and a larger G value indicates a greater change in the user’s position. α is the velocity correlation factor, and a larger α indicates a more random user’s movement. To verify the robustness of the algorithm, we chose relatively random user movement parameters; therefore, we set G=0.5 and α=0.7. $\bar{v}$ and $\bar{\theta }$ represent the historical average values of speed and steering angle.

### 2.2. UAV Dynamic Constraints

To ensure that a UAV can safely and stably complete its mission during flight, motion constraints based on dynamic limitations and environmental conditions need to be imposed. Because of their limited size, UAVs can carry a limited amount of fuel, and the time and distance required for flight tasks depend on the amount of fuel available. Therefore, flight distance constraints must be considered when planning a flight path.(3)$${fit}_{3}=\left\{\begin{array}{c} L_{i}-L_{max}\;\;\;\;L_{i}\geq L_{max}\\ 0{\;\;\;\;\;\;\;\;\;\;\;\;\;\;\;\;\;\;L}_{i}<L_{max} \end{array}\right.,$$
where $L_{max}$ represents the maximum flight distance of the UAV.

During the UAV flights, if the horizontal turning angle is excessively large, the UAV may overturn and crash. To prevent such accidents, constraints must be imposed on the turning angle of the UAV. A sharp turning angle helps maintain flight stability, particularly in complex environments such as those with high wind speeds. To render the turning angle constraint more prominent, we amplified the penalty by a factor of 15 through a proportional factor, ensuring that the penalty for the turning angle constraint is at the same level as that for the UAV’s maximum flight distance constraint.(4)$${fit}_{4}=\left\{\begin{array}{c} 15*\frac{{∆\theta }_{t}-\frac{\pi }{2}}{\pi }\;{\;\;\;\;\;∆\theta }_{t}\geq \frac{\pi }{2}\\ 0{\;\;\;\;\;\;\;\;\;\;\;\;\;\;\;\;\;\;\;∆\theta }_{t}<\frac{\pi }{2} \end{array}\right.,$$
where ${∆\theta }_{t}$ represents the change in the UAV’s turning angle at the current time step.

### 2.3. Environmental Threats

When planning a UAV’s flight path, it is essential to ensure that the UAV maintains a safe distance from obstacles in the environment. To simulate a real-world scenario, environmental threats were modeled as buildings and mountainous terrain, as shown in Figure 3. Buildings were represented as rectangular prisms, and mountainous terrain was represented using a two-dimensional Gaussian peak function.

The adaptation function ${fit}_{5}$ for the UAV and environmental threats can be expressed by Equation (5); the penalty increases as the UAV approaches a threat.(5)$${fit}_{5}=\left\{\begin{array}{c} 10*(1-\frac{d(x,y)}{d_{safe}})\;\;\;\;\;\;if\;d(x,y)\leq d_{safe}\\ 0\;\;\;\;\;\;\;\;\;\;\;\;\;\;\;\;\;\;\;\;\;\;\;\;\;\;\;\;\;\;\;\;\;\;\;\;\;if\;d(x,y)>d_{safe} \end{array}\right.,$$
where d(x,y) represents the distance from the UAV to the boundary of the obstacle, and $d_{safe}$ represents the minimum safe distance between the UAV and obstacle. To ensure consistency in the penalty values between the obstacle avoidance constraint and the first two constraints, we set the scaling factor to 10.

### 2.4. Joint Optimization Objective

The air-to-ground communication model between UAV and ground users differs from traditional ground communication. The propagation environments of air-to-ground (A-G) communication links are complex and variable. Under LOS conditions, signal propagation is relatively smooth, and path loss is minimal. However, under non-line-of-sight (NLOS) conditions, obstacles such as buildings and terrain cause significant signal attenuation. To better simulate the communication path between the UAV and users in a real environment, we employed the free-space model to characterize the signal propagation dynamics [20,21].

The path loss $PL(d_{k})$ between UAV i and user k can be expressed as:(6)$$PL(d_{k})=20\log\left(\frac{4\pi f_{c}d_{k}}{c}\right)+P(LOS)\eta_{LOS}+P(NLOS)\eta_{NLOS},$$
where $d_{k}$ represents the Euclidean distance between the UAV and the user, c is the speed of light, i.e., $3*10^{8}m/s$, and $f_{c}$ denotes carrier frequency. The values of these parameters are reported in Table 1. P(LOS) and P(NLOS) represent the LOS and NLOS probabilities, respectively, and $\eta_{LOS}$ and $\eta_{NLOS}$ represent the additional path loss in LOS and NLOS conditions, respectively.(7)$$P(LOS)=\frac{1}{1+a\exp\left(-b(\theta -a)\right)},$$(8)P(NLOS)=1−P(LOS),
parameters a and b depend on the environment, with a=20 and b=2; θ represents the elevation angle, i.e., the angle between the UAV and ground user.

The Signal-to-Noise Ratio (${SNR}_{i,k}$) can be expressed as(9)$${SNR}_{i,k}=\frac{P_{k}-PL(d_{k})}{N},$$
where $P_{k}$ represents the UAV transmission power, $P_{k}=P_{0}+PL(d_{k})$, $P_{0}$ represents the transmit power, and N  represents the noise power.

According to Shannon’s formula, the throughput $C_{i,j}$ between UAV i and user k can be expressed as:(10)$$C_{i,k}=B\;\log_{2}\left(1+\beta \;{SNR}_{i,k}\right),$$
where B represents the channel bandwidth. β is used to represent channel characteristics and can be adjusted according to different communication environments.

The total throughput for a single UAV m covering all users n can be expressed as:(11)$${fit}_{1}=\sum_{i=1}^{m}\sum_{k=1}^{n}B\;\log_{2}\left(1+\beta \;{SNR}_{i,k}\right)$$

The coverage fitness function can be expressed as the number of covered users:(12)$${fit}_{2}=L,$$
where L represents the number of covered users.

In this study, our goal was to jointly optimize the weight coefficients and learning factors in the PSO algorithm to maximize both user coverage and throughput. We propose the following joint optimization model to simultaneously optimize both coverage and throughput.

Finally, we combine the two objective functions, constraints, and environmental threats into a single fitness function using a weighted sum:(13)$${fit}_{tatol}=\tau_{1}\;{fit}_{1}+\tau_{2}\;{fit}_{2}-\tau_{3}{fit}_{3}-\tau_{4}{fit}_{4}-\tau_{5}{fit}_{5},$$
where $\tau_{i}$ represents the weight coefficient, which can be adjusted to prioritize different objectives based on actual requirements. If a particular objective is more critical in a specific application, its corresponding weight coefficient can be set higher, allowing the optimization result of that objective to carry greater significance in the overall fitness function, thereby influencing the final solution.

The aforementioned objective function and constraints are transformed into a nonlinear problem (NLP) as follows:(14)$$max{\;fit}_{total}\;\;for\;i=1,2,\ldots ,m\;s.t.\;L_{i}-L_{max}\leq 0\;{∆\theta }_{t}-\frac{\pi }{2}\leq 0\;\frac{d(x,y)}{d_{safe}}-1\leq 0$$

PSO may become trapped in local optima when addressing high-dimensional or complex nonlinear problems. Therefore, the NLP can be addressed by incorporating constraints into the objective function through an interior penalty function, which guides the PSO process and helps avoid infeasible solutions [22].(15)$${\;fit}_{total}=\tau_{1}\;{fit}_{1}+\tau_{2}\;{fit}_{2}-\sum_{i=1}^{m}\tau_{i}*P_{i}\;P_{3}=\max{\left(0,L_{i}-L_{max}\right)}^{2}\;P_{4}=\max{\left(0,{∆\theta }_{t}-\frac{\pi }{2}\right)}^{2}\;P_{5}=\max{\left(0,\frac{d(x,y)}{d_{safe}}-1\right)}^{2}$$
where $\max{\left(0,L_{i}-L_{max}\right)}^{2}$ represents the degree of violation when the flight distance exceeds the maximum limit. When the flight distance exceeds the maximum value, the penalty increases. $\max{\left(0,{∆\theta }_{t}-\frac{\pi }{2}\right)}^{2}$ represents the degree of violation when the turning angle exceeds the maximum limit. $\max{\left(0,\frac{d(x,y)}{d_{safe}}-1\right)}^{2}$ represents the degree of violation when the safe distance to the obstacle exceeds the maximum limit.

Using the interior penalty function method in PSO ensures that solutions remain within feasible constraints during the optimization process while still allowing the solutions to maximize the objective function.

### 2.5. Path Smoothing Algorithm

UAV paths often contain sharp turns or abrupt changes that do not comply with the drone’s heading-angle constraints. Therefore, in addition to path planning, it is necessary to smoothen the planned path.

The Catmull–Rom spline is a commonly used interpolation spline curve, characterized by its ability to generate smooth curves that pass through all control points. In Figure 4, the red path denotes before smoothing, and the blue path represents after smoothing. The Catmull–Rom spline is particularly suitable for scenarios that require paths to strictly cover the target points; it only changes the path around the deployment points without affecting the entire path. Therefore, the path can be adjusted locally to avoid obstacles and satisfy specific constraints. The smoothed curve generated by this method passes directly through all deployment points, ensuring that the smoothed path can still accomplish the task of covering dynamic users. Therefore, this study adopts the Catmull–Rom spline to smoothen the UAV paths.

## 3. Algorithm Improvement

In PSO, the initial particle positions affect convergence speed and quality [23]. Concentrated positions can cause inefficient exploration and local optima, whereas well-distributed positions help find the global optimum faster. Replacing random initialization with Tent mapping prevents particle concentration and improves the search diversity. The balance between local and global optima relies on inertia weight ω and acceleration factors $c_{1}$ and $c_{2}$, and incorrect settings lead to imbalance and premature convergence. Tent mapping dynamically adjusts these parameters by introducing randomness and chaos, which boosts early exploration and later-stage convergence.

### 3.1. Tent Mapping Initialization

Chaos refers to random, irregular motion within a deterministic system, characterized by unpredictability and uncertainty. Tent chaos is a piecewise linear mapping function commonly used to initialize the population state. With this method, the population state update follows a specific mapping formula:(16)$$z_{n+1}=\left\{\begin{array}{c} \mu *z_{n}\;\;\;\;\;\;\;\;\;\;\;\;\;\;\;\;if\;0<z_{n}\leq 0.5\\ \mu *\left(1-z_{n}\right)\;\;\;\;\;if\;0.5\leq z_{n}<1 \end{array}\right.,$$
where μ  is the chaotic coefficient, used to adjust the rate of state updates, and is typically set to 1.5. $z_{n}$ represents the state value of the system at the *n*-th iteration, and $z_{0}$ is a random number that prevents the algorithm from becoming stuck in a local optimum.

As μ increases, the system’s behavior transitions from a stable state to an unstable state, exhibiting bifurcations. As depicted on the right-hand side of Figure 5, the behavior becomes more complex, with multiple branches and chaotic regions. In this bifurcation diagram, the state variable $z_{n+1}$ presents a dense distribution of points across the entire range of μ, indicating that the Tent map sequence is evenly distributed across all possible states.

As reported in Figure 6, the values of the sequence appear to be distributed across the entire range, without clustering in any small part of the interval, and as the number of iterations increases, the Tent sequence is evenly distributed between [0, 1]. Therefore, the tent sequence may be ergodic because it covers different range intervals, which aligns with the characteristic of ergodicity.

First, the initial positions of the UAVs $x_{i}(0)\in \left[x_{max},x_{min}\right],y_{i}(0)\in \left[y_{max},y_{min}\right]$ are generated. The initial position of the UAV must be within a defined environmental range. For the x,y,z coordinates of each UAV, a chaotic sequence is generated using Tent mapping. By multiplying the Tent chaotic sequence by the UAV coordinates, the initial positions of the UAVs, which are evenly distributed within the search space, can be obtained.(17)$$x_{i}(0)=z_{0}*x_{max},y_{i}(0)=z_{0}*y_{max},$$(18)$$v_{i}(0)=z_{0}*v_{max},$$

### 3.2. Adaptive Inertia Weight Adjustment

In PSO, the inertia weight ω controls a particle’s dependence on its previous velocity during the update. An appropriate weight helps the particle explore the search space effectively.(19)$$\omega_{i}=\omega_{min}+(\omega_{max}-\omega_{min})\frac{1}{1+e^{-\left({iter}_{i}/{iter}_{max}\right)}},$$
where $\omega_{max}$ and $\omega_{min}$ are the initial and final values of the inertia weight, respectively. ${iter}_{max}$ represents the maximum number of iterations of the algorithm, and ${iter}_{i}$ denotes the current iteration number. If the particle swarm tends to converge, (i.e., ${iter}_{i}$ increases), the inertia weight decreases, enhancing the local exploitation in the early stages. Conversely, if the particle swarm dispersion is large, the inertia weight increases, strengthening global exploration in the later stages.

### 3.3. Learning Factor Adjustment

The learning factors were adjusted using a nonlinear decay method, which enhanced exploration in the early stages and improved convergence in the later stages.(20)$$c_{i1}=c_{1max}*e^{-\gamma *{iter}_{max}},$$(21)$$c_{i2}=c_{2max}*\left(1-e^{-\gamma *{iter}_{max}}\right),$$
where γ is the decay factor, which controls the rate of exponential decay and growth. $c_{1max}\;$ and $c_{2max}$ represent the maximum values of individual and social factors, respectively.

By setting the learning factors using a nonlinear decay approach, the algorithm can adjust the parameters more flexibly. As the iterations progress, the nonlinear decay of the learning factors causes the particle search process to gradually converge. In the early stages of the algorithm, larger learning factors allow the particles to broadly explore the search space, thereby increasing the global search capability of the algorithm. As the number of iterations increases, the gradual reduction in the learning factors strengthens the fine adjustments of the particles to the current solution, ensuring a balance between the global search and local optimization and improving the convergence speed.

### 3.4. UAV Speed and Position Update Formulas

In PSO, the UAV velocity update formula incorporates the inertia weight, individual best positions, and global best positions to adjust the flight speed and direction of the UAV dynamically. This process allows the UAV to maintain its motion along its current direction while adapting its path based on environmental changes. The position update formula uses the velocity update results to calculate the next UAV position. The UAVs’ next speed and position can be updated as(22)$$V_{i}(t+1)=\left\{\begin{array}{c} v_{xi}(t+1)=\omega_{i}{\;v}_{xi}(t)+c_{1}\;z_{n}\;(p_{besti}-x_{i}(t))+c_{2}\;z_{n}\;(g_{besti}-x_{i}(t))\\ v_{yi}(t+1)=\omega_{i}{\;v}_{yi}(t)+c_{1}\;z_{n}\;(p_{besti}-y_{i}(t))+c_{2}\;z_{n}\;(g_{besti}-y_{i}(t)) \end{array}\right.,$$(23)$$X_{i}(t+1)=\left\{\begin{array}{c} x_{i}(t+1)=x_{i}(t)+v_{xi}(t+1)\\ y_{i}(t+1)=y_{i}(t)+v_{yi}(t+1) \end{array}\right.$$
where $g_{besti}$ represents the global best value, and $p_{besti}$ represents the individual best value.

### 3.5. Tent–PSO Procedure

This study combined Tent mapping with PSO to solve the UAV path-planning problem to cover dynamic users. The algorithm first uses Tent mapping to initialize the UAV positions and then calculates the fitness of each particle using the objective function ${fit}_{total}$. The best solution is selected as the individual best position. Then, based on the fitness values and the improved $\omega_{i}$,$c_{i1}$,$c_{i2}$, the particle’s next position can be calculated. If this position causes a collision, the second-best solution is chosen instead, and the process continues until the maximum number of iterations is reached. The algorithm flowchart is shown in Figure 7.

The main procedures of Tent–PSO are presented in detail in Algorithm 1.
**Algorithm** **1** Tent–PSO1:**Input:** ${iter}_{max}=300$;2:**Output:** final UAV positions and paths;3:**for** each i
**do**4:$x_{i}(0)=z_{0}*x_{max},y_{i}(0)=z_{0}*y_{max},v_{i}(0)=z_{0}*v_{max}$;5:Evaluate i and set $p_{besti}=(x_{i}(0),y_{i}(0))$;6:$g_{besti}=\min\left[p_{besti}\right]$;7:**end** **for**8:**for** iter< ${iter}_{max}$
**do**9:    **for** each i
**do**10:      Update $\omega_{i}$ and $c_{i}$ by (19)–(21);11:      Update $v_{i}(t)$ and $x_{i}(t)$ by (22) and (23);12:      Update ${fit}_{total}$ by (13–15);13:        **if**
${fit}_{total}<{fit(p}_{besti})$14:            $p_{besti}=(x_{i}(i),y_{i}(t))$;15:        **else**16:            $p_{besti}=g_{besti}$;17:        **end** **if**18:    **end** **for**19:**end** **for**20:print $g_{best}$

## 4. Experimental Results

To verify the rationality of the proposed improved PSO, a simulation experiment on the UAV coverage for moving users was conducted using Python 3.7. To ensure the rigor of the meticulousness, we set obstacles in the environment with a specific distribution, using rectangular prisms to represent buildings and a 2D Gaussian function to represent mountainous terrain.

### 4.1. Simulation Parameter Design and UAV Path

The parameters used in the algorithm are listed in Table 1. We considered the impact of buildings and terrain, focused on how to cover dynamic users, and compared the results to those of the traditional PSO and PSO-GWO [24]. We conducted five comparative experiments and compared the performances of the three algorithms using the average values of the five experiments.

We assumed the presence of obstacles in the environment, such as buildings and terrain. UAVs must avoid these obstacles while providing coverage communication to ground users. Figure 8 shows a schematic of the UAV coverage of dynamic users. Users move randomly on the ground, and different UAVs are represented by different colors, with their communication ranges shown by the corresponding rings.

Figure 9 shows all the deployment points generated by the UAVs based on user movement. As users moved to the ground, the UAV deployment positions changed accordingly. After each iteration, the UAVs used the PSO algorithm to calculate the individual best value $p_{besti}$ and the global best value $g_{besti}$, which were then compared to the current fitness value to determine whether new coverage points needed to be generated.

Figure 10a shows the UAV path planning in a three-dimensional (3D) environment. For clarity, the paths were transformed into a two-dimensional plane. Figure 10b shows that the planned paths contain sharper turns because the Tent–PSO method does not include a path smoothing process. Consequently, the paths tend to have a more linear and segmented structure.

By contrast, Figure 10c shows the results obtained after path smoothing. It is evident that the smoothed UAV paths were more stable during flight, reducing unnecessary path oscillations and improving flight efficiency and safety. In addition, sharp turns in the original path introduced additional energy consumption to the UAV. The smoothed paths allowed the UAV to make more efficient use of its onboard energy while executing tasks, extending flight duration and increasing the task completion rate.

### 4.2. Comparison of Experimental Data

The proposed Tent–PSO was compared to the other two algorithms in terms of fitness function, throughput, and coverage. Figure 10 shows the variation curves of the average values for different algorithms as the number of iterations increases. As shown in Figure 11a, the fitness of all three algorithms increased with the number of iterations. After iteration, the fitness values for PSO-GWO and traditional PSO were 18.54 and 16.41, respectively, whereas Tent–PSO achieved a fitness of 31.79, far exceeding those of the other two algorithms. The figures show that PSO-GWO and PSO converged faster, indicating that these algorithms are likely to become stuck in local optima.

As shown in Figure 11b, a comparison of the throughput results of the three algorithms showed that the throughput for PSO-GWO and traditional PSO was 36.41 bps and 32.23 bps, respectively, while the highest throughput for the Tent–PSO algorithm was 62.69 bps, exceeding PSO-GWO by approximately 41.92%. Therefore, in dynamic environments, Tent–PSO can provide better communication quality for users.

A comparison of the average coverage rate indicators is shown in Figure 12. As shown in the figure, Tent–PSO consistently maintained a high coverage rate of to 100%. Although the ground users were always dynamically moving, causing fluctuations in UAV coverage, Tent–PSO maintained a dynamic user coverage rate of 89.76% after the iterative calculations. The average coverage rate for traditional PSO was only 58.23%, while PSO-GWO achieved an average coverage rate of 66.63%. This demonstrates that Tent–PSO has a strong adaptability to dynamic environments and can cover more communication users.

Tent–PSO leverages the chaotic characteristics of Tent mapping by introducing chaotic sequences to initialize particle positions. This approach improves the algorithm’s performance at the beginning of the iteration, offering a foundation for the subsequent search. However, PSO and PSO-GWO do not improve the particle position.

During the iterative process, the learning factors control the individual and global best values of the particles in the search space. The weight coefficients control the speed of particle movement during the search process. By adjusting the learning factors and weight coefficients using a nonlinear strategy, the algorithm gradually guides the particle swarm to converge to a global optimum. In the early stages of the algorithm, larger learning factors and weight coefficients work together to help the particles explore a broader search space. In the later stages, the nonlinear decay of the learning factors and weight coefficients allows all particles to focus on the optimal solution in the search space. Therefore, Tent–PSO achieved better solutions in later iterations.

Figure 13 and Figure 14 present the error analysis of the fluctuation range of the results for each algorithm across the five experiments. Figure 13 presents the error analysis results of fitness and throughput. Although Tent–PSO exhibited larger error fluctuations, it consistently maintained higher fitness and throughput values.

Figure 14 shows the error analysis of the coverage. Tent–PSO consistently maintained a higher coverage rate with smaller error fluctuations, indicating that it has better coverage stability in dynamic environments. The error analysis data for the three algorithms are summarized in Table 2.

## 5. Discussion

To address the dynamic user coverage communication problem, we introduced the Tent–PSO algorithm to plan a UAV’s path and maximize user coverage while maintaining the link quality. The Tent mapping used in the algorithm can adjust the UAV’s position and speed based on current environmental information. The adaptive weight coefficients and learning factors also help prevent the algorithm from falling into the local optima. In the same environment, with the Tent–PSO algorithm, the UAV coverage was maintained at 89%, with a throughput around 62.69 dps.

Simulation results show that in dynamic environments, Tent–PSO can not only maximize user coverage but also maintain the quality of the communication link between users and the UAV. And, in different user movement environments, the Tent–PSO can be used to calculate the best communication path between the relay UAV and users. When the height of UAVs increases, the corresponding coverage area will increase, but the communication quality will decrease. How to plan the 3D flight path of a UAV to balance the coverage area and communication quality is also an important part of this work. In the future, we can further study the deployment of the UAV 3D environment.

## Figures and Tables

**Figure 1 sensors-25-02005-f001:**
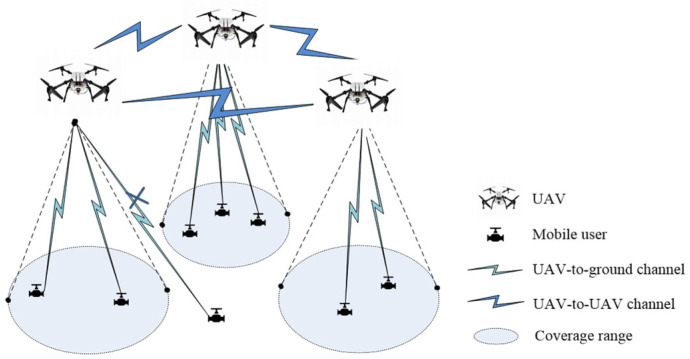
Environmental model.

**Figure 2 sensors-25-02005-f002:**
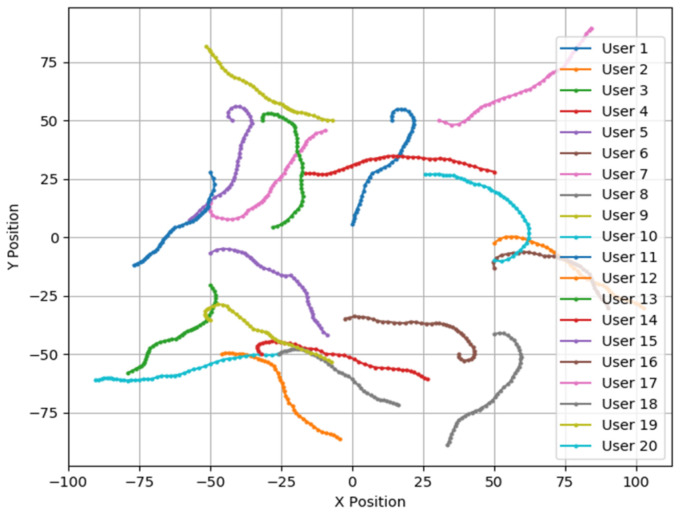
User mobility model.

**Figure 3 sensors-25-02005-f003:**
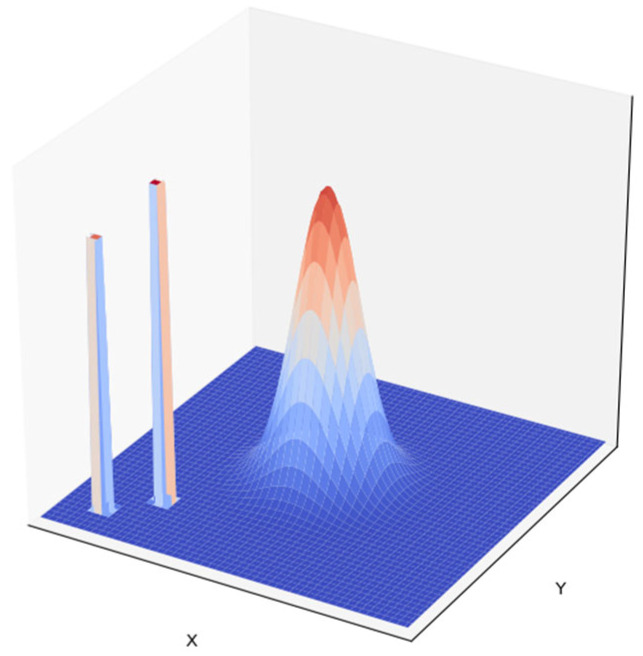
Environmental simulation diagram.

**Figure 4 sensors-25-02005-f004:**
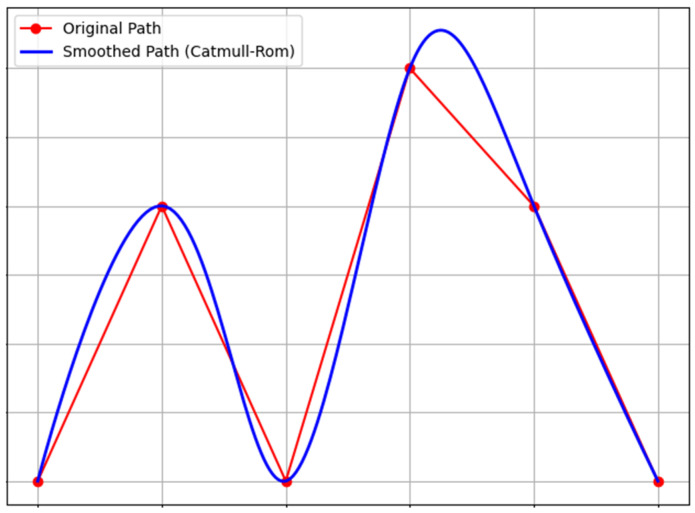
Catmull–Rom spline.

**Figure 5 sensors-25-02005-f005:**
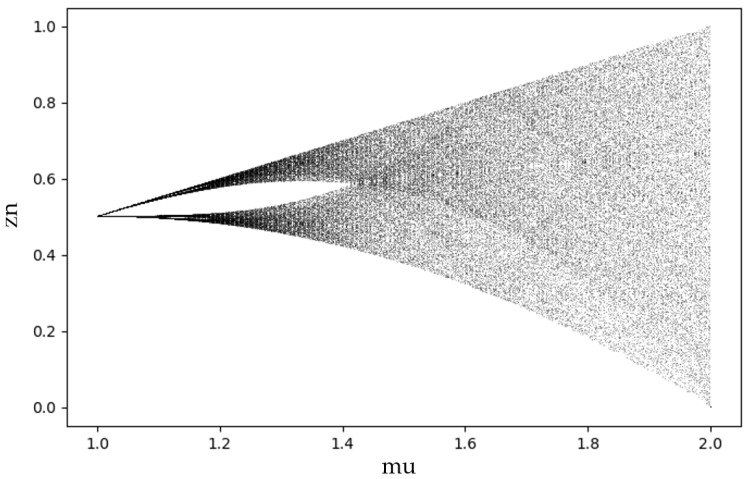
Tent mapping form.

**Figure 6 sensors-25-02005-f006:**
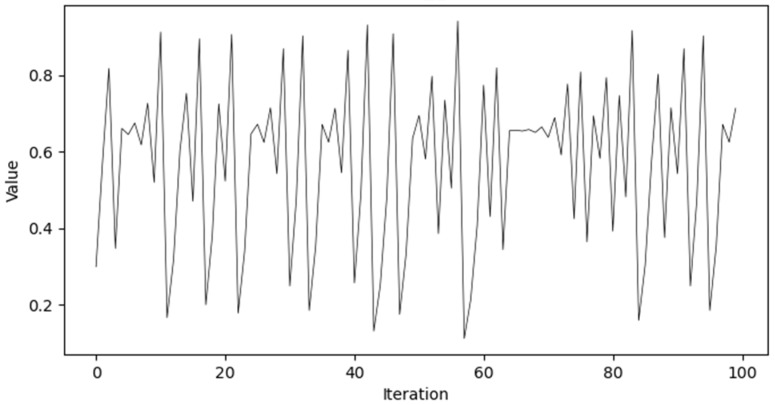
Tent mapping distribution plot.

**Figure 7 sensors-25-02005-f007:**
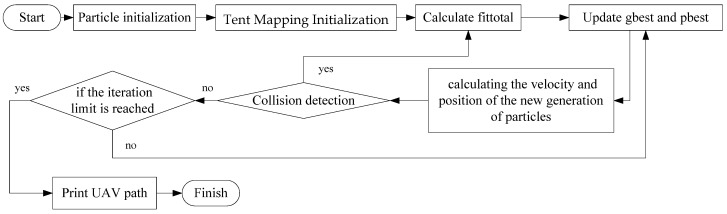
Tent–PSO procedure.

**Figure 8 sensors-25-02005-f008:**
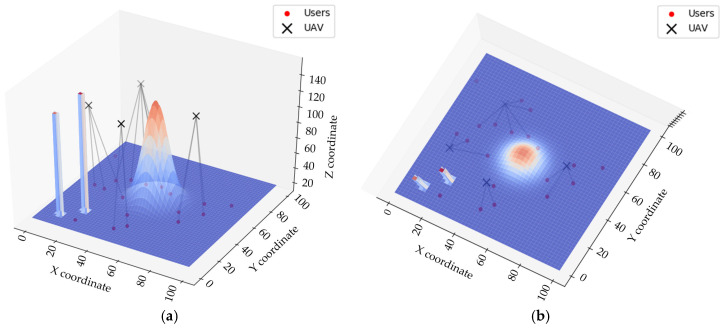
Illustration of UAV–user communication. (**a**) Front view; (**b**) top view.

**Figure 9 sensors-25-02005-f009:**
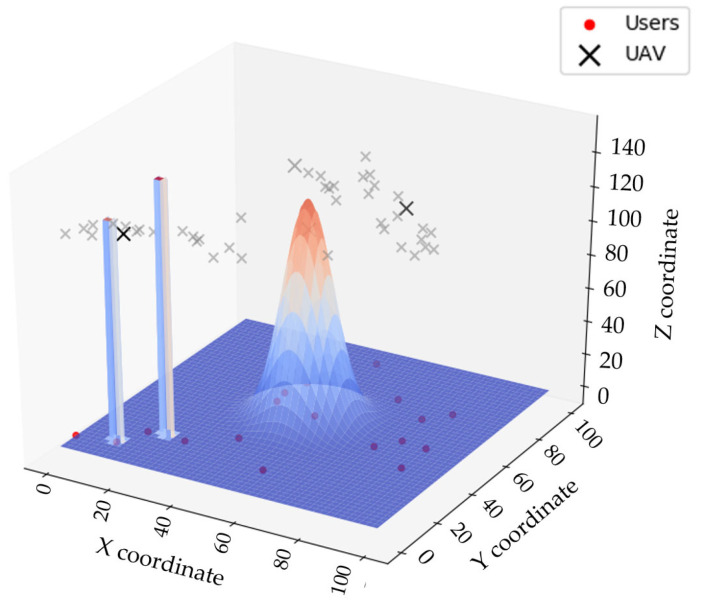
UAV deployment points.

**Figure 10 sensors-25-02005-f010:**
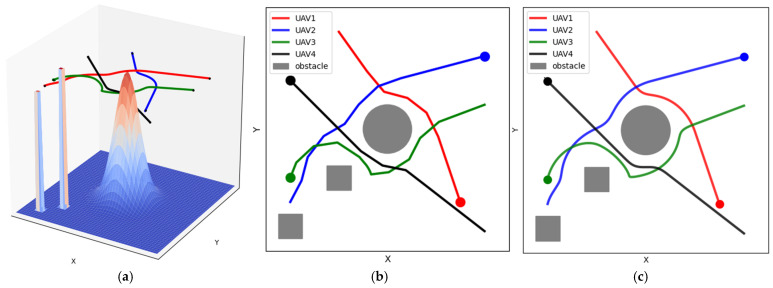
UAV path-planning results. (**a**) 3D path-planning results; (**b**) before path smoothing; (**c**) after path smoothing.

**Figure 11 sensors-25-02005-f011:**
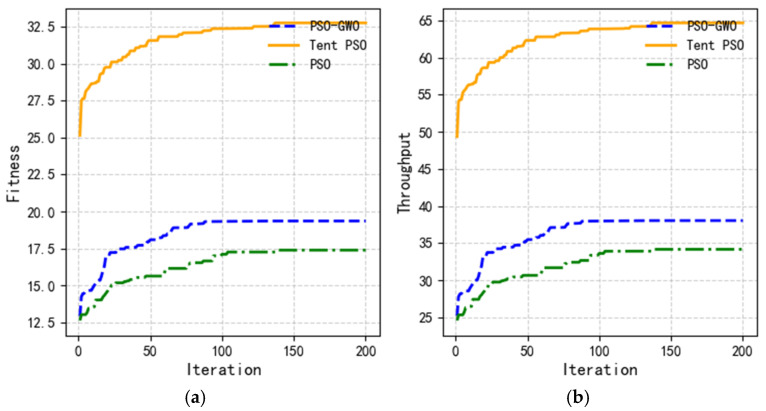
Variation curves of the average values for different algorithms. (**a**) Comparison based on fitness; (**b**) comparison based on throughput.

**Figure 12 sensors-25-02005-f012:**
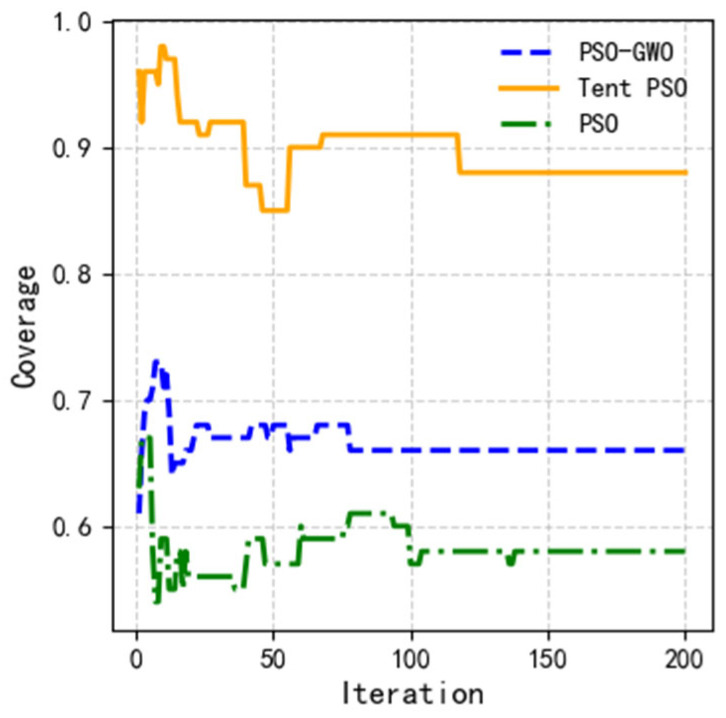
Variation curves of coverage for different algorithms.

**Figure 13 sensors-25-02005-f013:**
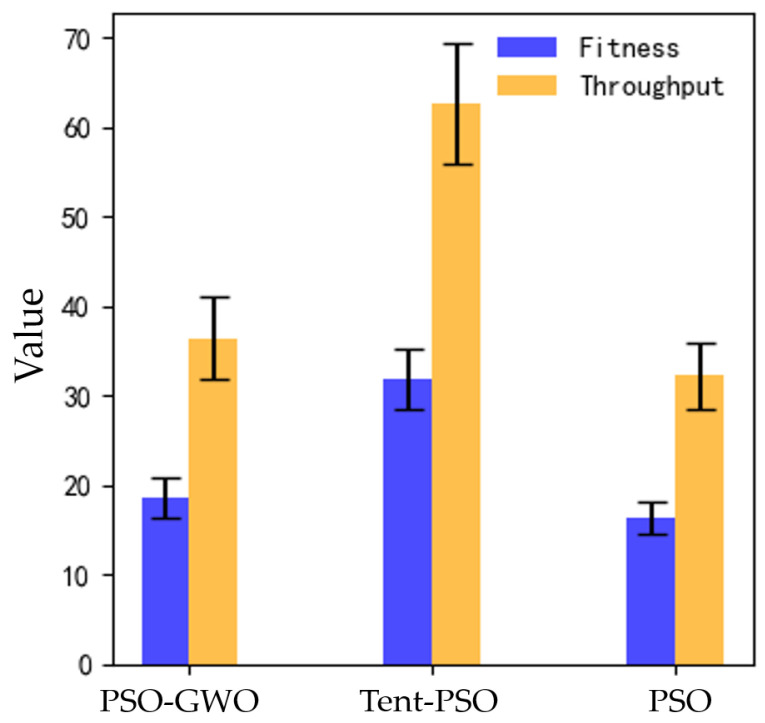
Comparison based on fitness and throughput.

**Figure 14 sensors-25-02005-f014:**
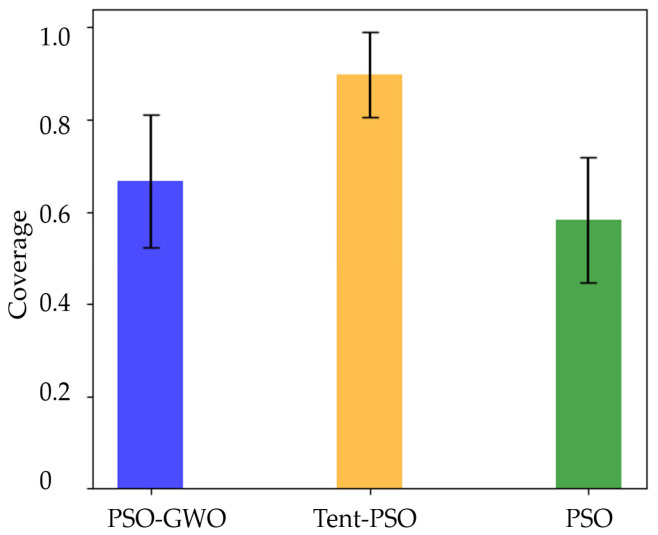
Comparison based on coverage.

**Table 1 sensors-25-02005-t001:** Parameter setting.

Parameter	Description	Value
i	UAV number	4
k	Users number	20
$d_{safe}$	UAV safe distance	10
$\eta_{los}$	path loss exponent under LOS	3
$\eta_{nlos}$	path loss exponent under NLOS	3.5
$\tau_{1}$	weight coefficient	0.3
$\tau_{2}$	weight coefficient	0.4
$\tau_{3}$	weight coefficient	0.1
$\tau_{4}$	weight coefficient	0.1
$\tau_{5}$	weight coefficient	0.1
γ	decay factor	0.9
$\omega_{max}$	initial value of the inertia weight	1.2
$\omega_{min}$	final value of the inertia weight	0.3
$P_{0}$	UAV’s transmission power	1 MHz
$f_{c}$	carrier frequency	2 GHz
N	noise power	−20 dBm
B	channel bandwidth	10 MHz
β	channel constant factor	1
μ	chaotic coefficient	0.5
$c_{1max}$	maximum value of individual factor	2.5
$c_{2max}$	maximum value of social factor	1

**Table 2 sensors-25-02005-t002:** Comparison of the performance of different algorithms.

	PSO	PSO-GWO	Tent–PSO
fitness	1.76	2.28	3.37
throughput	3.64	4.65	6.71
Coverage	0.12	0.16	0.09

## Data Availability

Data are available from the authors upon request.

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
