# Peer review of "Tent–PSO-Based Unmanned Aerial Vehicle Path Planning for Cooperative Relay Networks in Dynamic User Environments"

_sensors, 2025, doi:10.3390/s25072005_

Round 1

Reviewer 1 Report

Comments and Suggestions for Authors

This paper presents a Tent-PSO-based UAV path planning method for covering dynamic users while maintaining communication link quality. The introduction of Tent chaotic mapping to enhance UAV position initialization and dynamically adjust PSO parameters is well-justified, and the comparative analysis with traditional PSO and PSO-GWO demonstrates the effectiveness of the proposed approach. However, there are some aspects that require clarification and refinement before the paper can be considered for publication.

While the paper effectively models the optimization problem with a weighted cost function, it is unclear how hard constraints (such as flight distance limits, turn angle restrictions, and collision avoidance) are strictly enforced during optimization. Since a cost function only penalizes violations rather than strictly enforcing constraints, the proposed approach may still generate infeasible solutions in some cases.

A potential improvement could be adopting an interior penalty function approach, commonly used in NLP (Nonlinear Programming) methods, where constraints are incorporated into the optimization process in a way that guarantees feasibility at all iterations.

The authors should clarify whether their current implementation rejects infeasible solutions outright or penalizes them progressively.

Could the authors discuss alternative constraint-handling mechanisms in their encoding strategy, particularly in relation to maintaining feasible UAV trajectories under strict constraints?

The paper adopts the Gaussian-Markov mobility model to simulate user movements. While this model provides smooth transitions in user positions, it may not always accurately reflect real-world movement patterns in highly dynamic environments.

Have the authors considered alternative mobility models such as the Random Waypoint Model or Self-Similar Least Action Walk (SLAW) Model, which might better capture the behavior of urban mobile users?

The assumptions regarding user mobility should be explicitly discussed, particularly regarding how model parameters were chosen and whether sensitivity analysis was performed to evaluate how different mobility patterns affect UAV performance.

Minor Comments:

Grammar and Writing Issues

In the abstract:

"By adjusting the UAV's velocity and position using Tent chaotic mapping, the UAVs can be evenly distributed across the entire environment."

→ "By utilizing Tent chaotic mapping to adjust the UAVs' velocities and positions, they can be more evenly distributed across the environment." (Ensuring subject consistency and clarity).

In Section 1:

"The dynamic user changes cause the working area of the UAVs to show dynamic changes during the task execution process."

→ "The movement of dynamic users causes the UAVs’ operating area to continuously change throughout the task." (Avoid repetition of "dynamic" and improve readability).

In Section 3.4:

"We use the free-space model to describe the signal propagation characteristics[16,17]."

→ "We employ the free-space model to characterize the signal propagation dynamics."

The UAV flight constraints (e.g., distance, turning radius) are introduced in Section 3.2, but there is some inconsistency in how they are expressed. Some constraints use inequality notation, while others use a penalty function directly. For clarity, the notation should be consistent, and constraints should be formally defined before they are integrated into the fitness function.

Clarify how hard constraints (e.g., flight range, turning angle) are enforced in the algorithm—consider adopting interior penalty functions or constraint-handling strategies from NLP-baed methods (e.g., see 10.1109/TITS.2023.3316175).

Author Response

Thank you for your feedback on our manuscript. We have carefully revised the issues in the manuscript.

Reviewer 2 Report

Comments and Suggestions for Authors

The potential of Unmanned Aerial Vehicles (UAVs) as temporary communication relays has been well recognized. This paper investigates the challenges in current UAV path planning and proposes a novel path planning method for dynamic user scenarios. The application of Tent-PSO in UAV path planning demonstrates promising innovative value, especially in optimizing dynamic user coverage. However, the manuscript suffers from issues related to redundant language and grammatical errors, which negatively impact its readability and professionalism. It is recommended to enhance the discussion on the problem background, provide a more comprehensive review of existing methods, and clearly articulate the research objectives and contributions.

Major Concerns and Suggestions for Improvement:

  1. In section 1 Introduction, while the introduction highlights the advantages of UAVs as mobile communication relays, it lacks a concise overview of existing UAV path planning methods, particularly emphasizing why the limitations of traditional PSO need to be addressed. Recommendation: Add a paragraph summarizing existing UAV path planning methods (e.g., PSO and other heuristic approaches), highlighting their strengths and weaknesses, and emphasizing the innovations brought by Tent-PSO. Strengthen the discussion on the problem background, critique existing methods, and clearly define the research objectives and contributions.
  2. In Section 2 Related Works, although the section mentions how Deep Reinforcement Learning (DRL) addresses UAV communication interruptions, it fails to clearly discuss its advantages and disadvantages in path planning. Recommendation: Include a critical review of DRL methods and justify the choice of bio-inspired optimization algorithms in this study.
  3. The comparison of bio-inspired optimization algorithms is insufficient. While methods such as spiral UAV placement and simulated annealing are mentioned, their limitations are not discussed. Given that the core of this paper is the improvement of PSO, the section should provide a deeper analysis of existing PSO applications in UAV path planning, focusing on their challenges and shortcomings.
  4. Is Ref [11] based on a bio-inspired optimization algorithm for UAV path planning? What specific method does it use, and what are its characteristics?
  5. Provide a comparative analysis of Ref [11] and Ref [12] regarding their approaches to solving the issue of unnecessary energy and time consumption when establishing communication links between UAVs and users. Highlight their limitations and use this discussion to lead into the objectives of the current study.
  6. In Equation (1), the meanings of parameters G_θ, G_v, α, v\bar{v}, and θ\bar are unclear. Recommend clearly define these parameters and explain how they are calculated or set. In Equation (4), the parameter Δθ_t represents the turning angle change at the current time step. Explain why π/2 is used as the constraint and justify the choice of the scaling factor 5.
  7. In Equation (5): The parameter dsafe denotes the minimum safe distance. Specify how dsafe is determined and justify the scaling factor 10 used in the penalty function. In Equations (6) to (12), provide detailed explanations and clearly define all parameters used in these equations. Enhance the clarity and professionalism of the mathematical descriptions.
  8. On Figure 4, what is the frequency of the red points in Figure 4? Is the Catmull-Rom spline used for interpolation? Does the curvature of the interpolated spline adhere to the UAV's turning constraints? Recommendation: Provide a detailed explanation of the advantages of using Catmull-Rom splines for path smoothing and discuss how it ensures compliance with UAV maneuverability constraints.
  9. In Equation (14), variables zn and μ are undefined. Clearly define these variables and their roles in the Tent mapping process. The variable z0z_0 is undefined. Provide Equations (15) and (16) a definition and explain its significance.
  10. There may be an error in Equation (17), Review and correct the formula if necessary. Additionally, explain the choice of variables and parameters, including why itermax does not appear explicitly in the equation. Specify the values of ω when i=0 and i=itermax.
  11. In Equations (18) and (19), the learning factors c1c_1 and c2c_2 are adjusted using a nonlinear decay strategy. Explain how this adjustment improves the convergence of the algorithm and the rationale behind the chosen decay method.
  12. Although the improvements of Tent-PSO are described in detail, the paper lacks a comprehensive overview of the entire algorithm flow. Include a step-by-step description of the Tent-PSO workflow and discuss specific performance differences compared to traditional PSO.
  13. The description of environmental obstacles is insufficient (e.g., the number of obstacles, distribution patterns, etc.). Provide a detailed description of the experimental scenario, including all relevant parameters and settings.
  14. While coverage rate, throughput, and fitness are mentioned as evaluation metrics, their calculation methods are not explicitly defined. Clearly define how each metric is computed. Additionally, provide a more thorough discussion on why Tent-PSO outperforms the other algorithms, focusing on specific factors contributing to its superior performance.
  15. The paper introduces a potentially valuable approach (Tent-PSO) for UAV path planning in dynamic user scenarios. However, significant revisions are needed to improve the clarity, completeness, and professionalism of the manuscript. Addressing the above concerns will strengthen the paper's contribution and make it suitable for publication.
Comments on the Quality of English Language
  1. Some sentences are overly lengthy and contain repetitive expressions.
  2. Inconsistent terminology usage, such as alternating between "UAV" and "relay UAV," should be standardized. The phrase "dynamic user changes" is unnatural; suggested revisions include "dynamic user mobility" or "movement of dynamic users."
  3. Incorrect article or word form usage: "a neural network-based strategy is proposed [8,9]" should be revised to "has been proposed" to maintain tense consistency.

Author Response

(The authors gave the same response as above.)

Reviewer 3 Report

Comments and Suggestions for Authors

The manuscprit presents an improved UAV path planning approach using a Tent-PSO algorithm to optimize coverage and throughput for dynamic user environments. Hence, the following issues needs an improvement:

  1. The title seems unclear and better to refine it as "Tent-PSO-Based UAV Path Planning for Cooperative Relay Networks in Dynamic User Environments."
  2. Abstract: Consider to reconstruct your abstract by considering background, problem statement, methods and your results.
  3. Introduction: it is expected to show the motivation, research gap and contributions  more explicitly. Moreover, why existing PSO-based methods fail and how Tent chaotic mapping improves the situation should be clearly analyzed.
  4. Figures & Diagrams: The study may illustrative diagram explaining how Tent mapping initializes UAV positions and improves PSO behavior. Moreover, A flowchart of the Tent-PSO process would be suggested to easily follow the algorithm more easily. Mathematical Notation: Some equations lack clear explanations of their parameters.
  5. Methodology: Some graphs lack proper axis labels and legends (e.g., Figures 8 & 9), Consider using error bars in performance metrics to show result variability, A bar chart or table summarizing improvements (coverage, throughput, convergence speed) would be useful.
  6. Writing style and Grammar: Some sentences are overly complex and better to Simplify where possible. Avoid Redundancy: Some concepts (PSO limitations, Tent mapping benefits) are repeated multiple times.
  7. Generally, it is important to improve writing clarity, remove redundant descriptions, more figures/flowcharts etc. 
Comments on the Quality of English Language

it is suggested to Improve writing style and grammar.

Author Response

(The authors gave the same response as above.)

Round 2

Reviewer 1 Report

Comments and Suggestions for Authors

This paper should be further modified to improve its figure quality before it gets accepted. The authors have resolved the raised issues.

Author Response

Dear reviewer:

Thank you for your suggestions. 
Please find the attachment the revised file.

Reviewer 2 Report

Comments and Suggestions for Authors

The manuscript has been revised in accordance with the reviewers' comments and is recommended for acceptance. However, prior to publication, further refinements are suggested to strengthen the discussion on the applicability and limitations of the research methodology and to improve the organization of the manuscript. Some sections remain overly lengthy, particularly in the algorithm description, and should be divided into shorter segments to enhance readability and clarity.

Author Response

(The authors gave the same response as above.)

Reviewer 3 Report

Comments and Suggestions for Authors

N/A

Author Response

Dear reviewer:

Thank you for your valuable feedback, which has improved the revised manuscript.
Please find the attachment the revised file.
